# The hierarchical organization of the lateral prefrontal cortex

Derek Evan Nee[1,2]*, Mark D'Esposito[1,2]

[1]Helen Wills Neuroscience Institute, University of California, Berkeley, United States; [2]Department of Psychology, University of California, Berkeley, United States

**Abstract** Higher-level cognition depends on the lateral prefrontal cortex (LPFC), but its functional organization has remained elusive. An influential proposal is that the LPFC is organized hierarchically whereby progressively rostral areas of the LPFC process/represent increasingly abstract information facilitating efficient and flexible cognition. However, support for this theory has been limited. Here, human fMRI data revealed rostral/caudal gradients of abstraction in the LPFC. Dynamic causal modeling revealed asymmetrical LPFC interactions indicative of hierarchical processing. Contrary to dominant assumptions, the relative strength of efferent versus afferent connections positioned mid LPFC as the apex of the hierarchy. Furthermore, cognitive demands induced connectivity modulations towards mid LPFC consistent with a role in integrating information for control operations. Moreover, the strengths of these dynamics were related to trait-measured higher-level cognitive ability. Collectively, these results suggest that the LPFC is hierarchically organized with the mid LPFC positioned to synthesize abstract and concrete information to control behavior.

*For correspondence: denee@berkeley.edu

**Competing interests:** The authors declare that no competing interests exist.

## Introduction

The prefrontal cortex (PFC) is central to higher-level cognition. Damage to the PFC produces severe deficits in goal-directed cognition (*Luria, 1966*; *Damasio and Anderson, 1993*; *Lhermitte, 1986*) and neuroimaging studies have demonstrated the involvement of the PFC across a wide range of tasks (*Duncan and Owen, 2000*; *Niendam et al., 2012*). Despite its ubiquitous influence, mechanistic insight into PFC function has remained difficult.

Progress towards understanding PFC function has been made through the development of organizing principles that describe gradients of function across the full extent of PFC. A prominent organizing principle is that the rostral/caudal axis of the lateral PFC (LPFC) is structured according to the level of cognitive control (*Fuster, 1997*; *Koechlin et al., 2003*; *Koechlin and Summerfield, 2007*; *Badre, 2008*; *Badre and D'Esposito, 2007*; *2009*). Mounting evidence from neuroimaging and lesion studies suggests that progressively rostral areas perform progressively higher levels of cognitive control through processing increasingly abstract representations (*Koechlin et al., 2003*; *Badre, 2008*; *Badre and D'Esposito, 2007*; *2009*; *Koechlin and Jubault, 2006*; *Nee and Brown, 2012*; *2013*; *Koechlin et al., 1999*; *Bahlmann et al., 2014*; *Azuar et al., 2014*). This organization enables the LPFC to control behavior across multiple timescales including adapting actions according to the prevailing context, and aligning behaviors with longer-term goals and plans (*Fuster, 2001*; *2008*).

While differentiating the functional sensitivities of sub-regions of the LPFC is an important first step, mechanistic insight into how the LPFC supports goal-directed cognition requires understanding intercommunication among LPFC sub-regions. To this end, it has been proposed that the LPFC is organized hierarchically such that more rostral areas exert asymmetrically greater influence upon more caudal areas (*Badre and D'Esposito, 2009*). Hierarchical processing confers substantial

**eLife digest** Part of the brain called the lateral prefrontal cortex has a critical role in many of the processes seen as hallmarks of human cognition, such as reasoning, planning and problem-solving. Individuals with damage to the lateral prefrontal cortex are disorganized and easily distracted, and may show behaviors that are inappropriate for their context. However, the involvement of the lateral prefrontal cortex in such a wide range of processes makes it difficult to study. This in turn presents a significant roadblock to a full understanding of cognition and human intelligence.

Of particular interest is whether the lateral prefrontal cortex has a hierarchical organization wherein a region coordinates the roles of other regions, much like the chief executive of a company. Therefore, Nee and D'Esposito set out to map how the lateral prefrontal cortex is organized, and how its different regions communicate with each other to support complex cognition.

Brain imaging revealed that the rear (posterior) part of the lateral prefrontal cortex processes an individual's current situation, while the front (anterior) prepares for future situations. Areas in the middle process both types of information. These central areas appear to be highly influential as they have stronger connections to the anterior and posterior regions than vice versa.

In cognitively demanding situations, the middle areas receive input from both anterior regions (potentially about future needs) and posterior regions (potentially about current needs). By combining the two sets of information, the middle areas can select behaviors that take into account both present circumstances and longer-term goals. With this strategic overview, the middle areas of the lateral prefrontal cortex are well positioned to play the part of the brain's chief executive.

Future experiments should test whether the interactions observed between the different regions of the lateral prefrontal cortex are essential for complex planning and thinking. Additional work in animals would improve our understanding of the mechanisms underlying these interactions. Finally, studying how these interactions are altered in disorders such as schizophrenia, where the lateral prefrontal cortex shows abnormal activity, might pave the way for more effective treatments.

computational efficiency and flexibility (*Botvinick and Weinstein, 2014*; *Duncan, 2013*; *Solway et al., 2014*), and such processing in the LPFC could explain the massive propensity for higher-level cognition found in primates. However, data demonstrating functional hierarchical processing in the LPFC have been limited (*Koechlin et al., 2003*) and controversial (*Reynolds et al., 2012*; *Crittenden and Duncan, 2014*). Furthermore, no data have linked hierarchical processing to its surmised cognitive advantages. Finally, although it has been widely assumed that the rostral-most areas of LPFC are situated at the apex of the hierarchy (*Koechlin and Summerfield, 2007*; *Badre and D'Esposito, 2009*), recent data call this assumption into question (*Goulas et al., 2014*). Hence, understanding whether and how hierarchical processing occurs in the LPFC remains a significant barrier to modeling goal-directed cognition.

Here, we examine how interactions within the LPFC support higher-level cognition. To do so, we devised a paradigm to engage the entire LPFC by orthogonally manipulating demands on stimulus domain and cognitive control (*Figure 1*). We began by confirming that the rostral/caudal axis of the LPFC operates according to abstraction principles by examining gradients of stimulus domain-generality/specificity, and temporal control over action. Next, we modeled interactions among LPFC sub-regions to examine whether and how cognitive control demands elicit asymmetrical influences indicative of hierarchical processing. Finally, we aimed to establish whether individual differences in hierarchical interactions predict trait-measured higher-level cognitive functioning, consistent with the presumed computational advantage of hierarchical processing.

## Results

To engage the full extent of LPFC necessary to test our predictions, we adapted a task (*Koechlin et al., 1999*; *Charron and Koechlin, 2010*) that orthogonally manipulated demands on stimulus domain (i.e. verbal vs. spatial) and cognitive control (*Figure 1*). Participants began each block of trials by performing a basic sequencing task which required the determination of whether the current stimulus followed the previous stimulus in a trace of the points of a star (spatial task) or

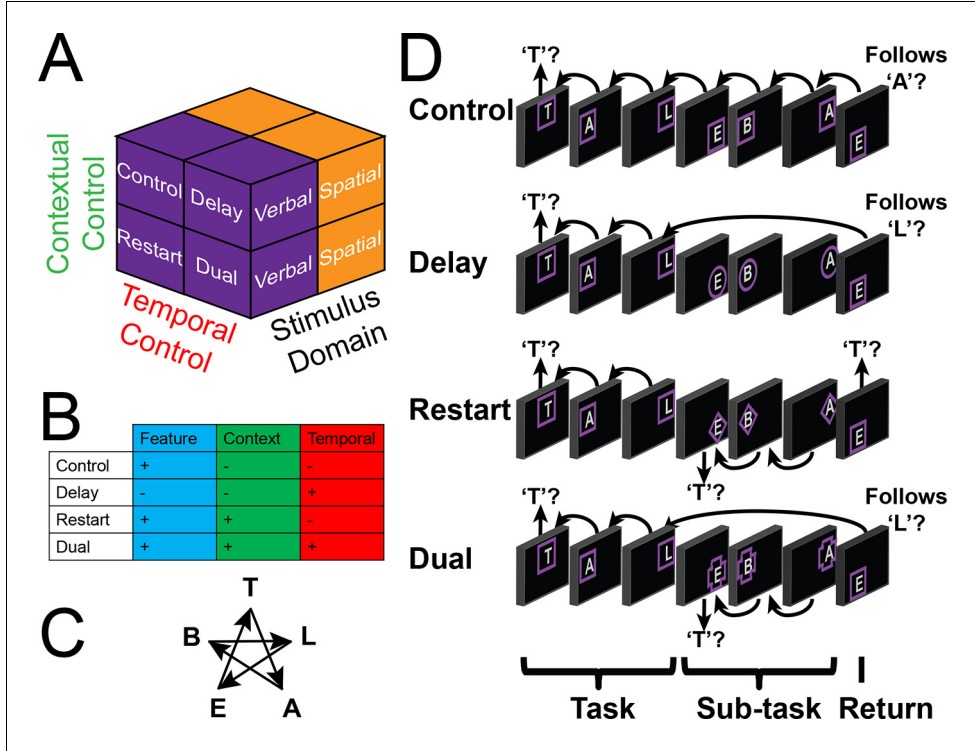

**Figure 1.** Experimental design. (A) The design orthogonally manipulated factors of *Stimulus Domain* (verbal, spatial) and two forms of cognitive control: *Contextual Control* (low – *Control*, *Delay*; high – *Restart*, *Dual*) and *Temporal Control* (low – *Control*, *Restart*; high – *Delay*, *Dual*). These factors were fully crossed in a 2 x 2 x 2 design. (B) Cognitive control processes engaged by condition. (C) The basic task required participants to judge whether a stimulus followed the previous stimulus in a sequence. The sequence in the verbal task was the order of the letters in the word 'tablet.' The sequence in the spatial task was a trace of the points of a star. The start of each sequence began with a decision regarding whether the currently viewed stimulus is the start of the sequence (e.g. 't' in the verbal task). (D) Factors were blocked with each block containing a basic task phase, a sub-task phase, a return trial, and a second basic task phase (not depicted), for all but the *Control* blocks. *Control* blocks consisted only of the basic task phase extended to match the other conditions in duration. Colored frames indicated whether letters or locations were relevant for the block (verbal – purple; spatial – orange in this example; verbal condition depicted). The basic task was cued by square frames. Other frames cued the different sub-task conditions. In the *Delay* condition (circle frames), participants held in mind the place in the sequence across a distractor-filled delay. In the *Restart* condition (diamond frames), participants started a new sequence. In the *Dual* condition (cross frames), participants simultaneously started a new sequence, and maintained the place in the previous sequence.

The following figure supplement is available for figure 1:

**Figure supplement 1.** Spatial task.

in the word 'tablet' (verbal task) with the appropriate stimulus feature cued by color. Hence, a critical aspect of the basic task was *Feature Control* – selecting the appropriate stimulus feature to integrate into ongoing cognition. These demands recruit caudal dorsal and ventral LPFC for spatial and verbal selection, respectively (*Nee et al., 2013*). Since there was no preceding stimulus at the start of each block, participants decided whether the first stimulus of a given block was the start of the sequence (i.e. top of the star for the spatial task, 't' for the verbal task).

After 2 to 5 trials, participants were cued to either perform one of three different sub-tasks (*Restart*, *Delay*, or *Dual*), or to continue with the basic task (*Control*). Tasks were cued by the shapes of frames with the basic/*Control* task always cued by square frames. After a 3 to 5 trial sub-task phase, participants were cued to return to the basic task, which we refer to as a return trial. Participants began the *Restart* task by starting the sequence anew, deciding whether the stimulus was the

start of the respective sequence, and then performing the basic task for the remainder of the sub-task phase. When cued to return the basic task, participants once again started the sequence anew. Thus, the *Restart* task required *Contextual Control* – selecting the appropriate action rules (i.e. sequence-match vs sequence-start) for the given circumstance. This process is involved, for example, in preparing to switch tasks (*Monsell, 2003*), comprising readying a new rule set as distinct from a previous one. Previous work has demonstrated that this type of control process engages the mid LPFC (*MacDonald et al., 2000*; *Sohn et al., 2000*).

In the *Delay* task, participants kept in mind their place in the sequence across a distractor-filled delay. During the delay, participants rejected all stimuli with a keypress corresponding to a 'no' response. When cued to return to the basic task, participants evaluated the current stimulus with respect to the delay-maintained memorandum. As a result, the *Delay* task placed minimal demands on current stimulus processing, while requiring participants to prepare to integrate past information into future processing thereby requiring *Temporal Control*. This is an important component of working memory, supporting the ability to maintain information in the face of ongoing processing in order to inform future cognition (*D'Esposito and Postle, 2015*; *Baddeley, 2003*). Previous work has demonstrated that this type of control process engages rostral LPFC (*De Pisapia et al., 2012*; *De Pisapia et al., 2007*; *Nee et al., 2014*).

Finally, the *Dual* task contained the essential processes of both the *Restart* and *Delay* tasks. Here, participants started the sequence anew, while simultaneously remembering their place in the previous sequence. When cued to return to the basic task, participants disengaged from the current sequence and evaluated the current stimulus with respect to the delay-maintained memorandum. Hence, this task required a combination of *Contextual Control* and *Temporal Control* to select the appropriate action rules and temporal origin of information (i.e. sequence-match current sequence vs sequence-match previous sequence).

Collectively, the task was a 2 x 2 x 2 design with factors of *Stimulus Domain* (verbal, spatial), *Contextual Control* (low – *Control*, *Delay*; high – *Restart*, *Dual*), and *Temporal Control* (low – *Control*, *Restart*; high – *Delay*, *Dual*). Since the *Delay* task required negligible demands on *Feature Control* due to the minimization of current processing, it was expected that the statistical interaction between the cognitive control main effects would reveal demands on *Feature Control*. Collectively, these manipulations were designed to engage the entire LPFC, providing an opportunity to examine the hypothesized hierarchical interactions among LPFC areas that support cognitive control. We focus our analyses on the sub-task trials, unless otherwise specified.

## Behavioral results

To examine behavioral manifestation of cognitive control demands, separate 2 x 2 x 2 ANOVAs were computed on error-rate (ER), and reaction times (RT) on correct trials (*Figure 2*). These analyses revealed significant effects of cognitive control demands with main effects of *Temporal Control* (ER: $F(1,23) = 5.72$, $p<0.05$; RT: $F(1,23) = 28.72$, $p<0.0001$), *Contextual Control* (ER: $F(1,23) = 16.21$, $p=0.0005$; RT: $F(1,23) = 147.79$, $p<10^{-11}$), and their interaction (ER: $F(1,23) = 23.78$, $p=0.0001$; RT: $F(1,23) = 87.16$, $p<10^{-8}$). As anticipated, the interaction was largely driven by fast and accurate performance on the *Delay* condition, which placed minimal demands on current processing during the sub-task phase. By contrast, there were no main effects of *Stimulus Domain* or interactions of *Stimulus Domain* with cognitive control demands (ER and RT all $p>0.05$).

Following each sub-task, there was a return trial that required either the integration of an item held across the sub-task (*Delay* and *Dual* conditions) or a switch to a new sequence (*Restart* and *Dual* conditions). These were analyzed with separate 2 x 2 x 2 ANOVAs on ER and RTs on correct trials (*Figure 2*). These analyses revealed a significant effects of cognitive control demands with main effects of *Temporal Control* in ER, but not RT (ER: $F(1,23) = 11.44$, $p<0.005$; RT: $F(1,23) = 1.89$, $p>0.15$), *Contextual Control* (ER: $F(1,23) = 20.00$, $p<0.0005$; RT: $F(1,23) = 87.05$, $p<10^{-8}$), and their interaction (ER: $F(1,23) = 19.18$, $p<0.0005$; RT: $F(1,23) = 10.24$, $p<0.005$). Again, no effects of *Stimulus Domain* were observed (ER and RT all $p>0.05$). These analyses confirm the effective manipulation of cognitive control demands, but equivalent loadings of control demands across stimulus domain.

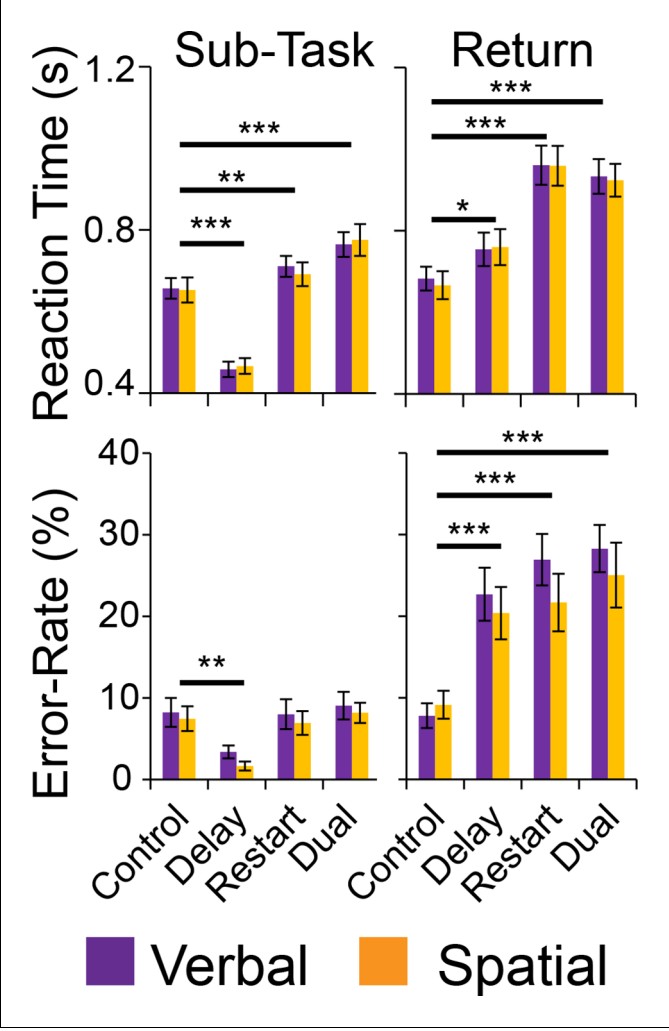

**Figure 2.** Behavioral results. Reaction times (top) and error-rates (bottom) as a function of condition in the sub-task (left) and return trials (right). Comparisons reflect Bonferroni corrected tests against the *Control* condition. *$p_{corrected}<0.05$, **$p_{corrected}<0.005$; ***$p_{corrected}<0.0005$.

## Evaluating the rostral/caudal axis and cognitive control

To verify the expected rostral/caudal gradient by cognitive control in the LPFC, fMRI data from the epoch of the sub-tasks (see Materials and methods for full modeling details) were analyzed with a 2 x 2 x 2 ANOVA. Here, we focus on the effects of cognitive control (*Figure 3*), while the effects of *Stimulus Domain* are depicted in *Figure 3—figure supplement 1*. The main effect of *Temporal Control* produced activations across the rostral-most aspects of the LPFC, mostly within the lateral frontopolar cortex (FPl; putative area 10) and mid-middle frontal gyrus (MFG; putative area 9/46), also spanning orbitofrontal cortex and pars orbitalis ventrally, through rostral-MFG (rMFG; putative area 46) laterally, up to the superior frontal gyrus dorsally. Previous studies have suggested that rostral LPFC is engaged as difficulty is increased regardless of functional demands (*Crittenden and Duncan, 2014*; *Fedorenko et al., 2013*). Notably, the main effect of *Temporal Control* in the present design includes both the easiest condition (*Delay*) and hardest condition (*Dual*). The rostral LPFC was engaged considering each condition in isolation (*Figure 3—figure supplement 2*), ruling out the possibility that activations in this area are attributable to difficulty. The main effect of *Contextual Control* was most pronounced in mid-lateral areas including the inferior frontal sulcus (IFS; putative area 45 or 9/46v) extending into pars triangularis ventrally, and the caudal middle frontal gyrus (cMFG; putative area 8Ar or 9/46d) dorsally. Significant effects extended both rostrally into the FPl,

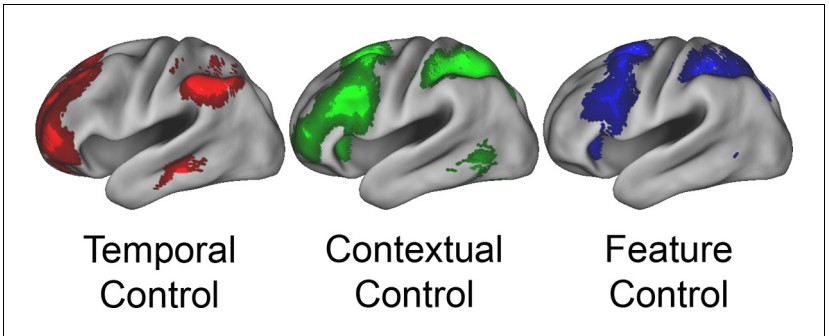

**Figure 3.** Univariate whole-brain results. Temporal demands of cognitive control produced a gradient of activation along the rostral/caudal axis of the LPFC. *Temporal Control* activated the rostral LPFC (red), *Contextual Control* the mid LPFC (green), and the interaction relating to *Feature Control* the caudal LPFC (blue). All results are corrected for multiple comparisons. Darker colors: $p<10^{-3}$; Brighter colors: $p<10^{-8}$ (*Temporal Control, Feature Control*) or $p<10^{-12}$ (*Contextual Control*) to better visualize peaks.

The following source data and figure supplements are available for figure 3:

**Source data 1.** Statistical parametric maps of the univariate whole-brain results.
**Figure supplement 1.** Univariate whole-brain results of *Stimulus Domain*.
**Figure supplement 2.** Effects of delay and dual.
**Figure supplement 3.** Activation magnitudes within regions-of-interest.
**Figure supplement 4.** Gradient visualization.
**Figure supplement 5.** Univariate whole-brain robustness.

and caudally into the inferior frontal junction (IFJ; putative area 44 or 8Av) and pars opercularis ventrally, and the caudal superior frontal sulcus (SFS; putative area 8Ad) dorsally. Finally, the statistical interaction of the effects of cognitive control produced the strongest foci in caudal LPFC areas including the IFJ and pars opercularis ventrally, and the caudal SFS dorsally, spreading into the mid-lateral PFC. As indicated in *Figure 3—figure supplement 3*, this interaction was primarily due to reduced activation during the *Delay* task, which had minimal demands on current processing. Thus, the effects are consistent with *Feature Control*. Collectively, these results demonstrate a rostral/caudal gradient as a function of cognitive control processes with an orderly progression of *Temporal Control, Contextual Control*, and *Feature Control* running from rostral to caudal spanning the entire LPFC (*Figure 3*; *Figure 3—figure supplement 4*). These effects were replicable when the dataset was split into two, confirming their robustness (*Figure 3—figure supplement 5*).

## Abstraction: stimulus domain-sensitivity and generality

The previous analyses demonstrated the predicted rostral/caudal functional gradient as a function of cognitive control processes. Next, we sought to verify that this gradient reflected increasingly abstract processing in increasingly rostral areas. If processing in the LPFC becomes more abstract in progressively rostral areas, caudal areas should process concrete features such as stimulus-domain, whereas more rostral areas should instantiate control processes that act across domains (e.g. a plan that can be applied to spatial or verbal stimuli). Thus, we reasoned that abstract versus concrete processing would reveal itself through stimulus domain-generality versus specificity.

To test this hypothesis, we used activation peaks of the LPFC ROIs defined by the cognitive control contrasts above. Notably, each contrast produced a distinct dorsal and ventral activation peak spanning rostral areas (FPl, MFG), mid areas (IFS, cMFG), and caudal areas (IFJ, SFS). Hence, this analysis enabled the examination of whether stimulus domain-sensitivity is present throughout

the rostral/caudal axis, or whether domain-generality emerges at a particular rostra/caudal level. First, we performed a 2 (dorsal, ventral) x 3 (rostral, mid, caudal) ANOVA on the main effect of *Stimulus Domain*. This analysis revealed a significant effect of the dorsal/ventral axis ($F_{(1,23)}$ = 100.21, $p<10^{-9}$), a significant effect of the rostral/caudal axis ($F_{(2,46)}$ = 10.63, $p<0.0005$), and critically, an interaction between the dorsal/ventral and rostral/caudal axes ($F_{(2,46)}$ = 56.12, $p<10^{-12}$). Follow-up t-tests revealed that caudal and mid ventral areas demonstrated verbal-sensitivity (SFS: $t_{(23)}$ = 5.93, $p_{corrected}<0.0001$; cMFG: $t_{(23)}$ = 5.08, $p_{corrected}<0.0005$), but not rostral areas (MFG: $t_{(23)}$ = 1.54, $p_{corrected}>0.8$). Symmetrically, caudal and mid dorsal areas demonstrated spatial-sensitivity (IFJ: $t_{(23)}$ = 7.83, $p_{corrected}<10^{-6}$; IFS: $t_{(23)}$ = 5.33, $p_{corrected}<0.0005$), but not rostral areas (FPl: $t_{(23)}$ = −2.25, $p_{corrected}>0.2$). A similar lack of stimulus domain-sensitivity was observed in a third rostral peak (rMFG: $t_{(23)}$ = 0.83, $p>0.4$). This region is not considered further due to lack of resting connectivity with the other LPFC regions (see Materials and methods; *Figure 6—figure supplement 4*). Collectively, these data indicate caudal and mid LPFC are sensitive to stimulus domain, whereas rostral LPFC areas are domain-general, consistent with a gradient of abstraction (*Figure 4*). These effects were replicable when the dataset was split into two, confirming their robustness (*Figure 4—figure supplement 1*). Given the sensitivity of caudal areas to stimulus

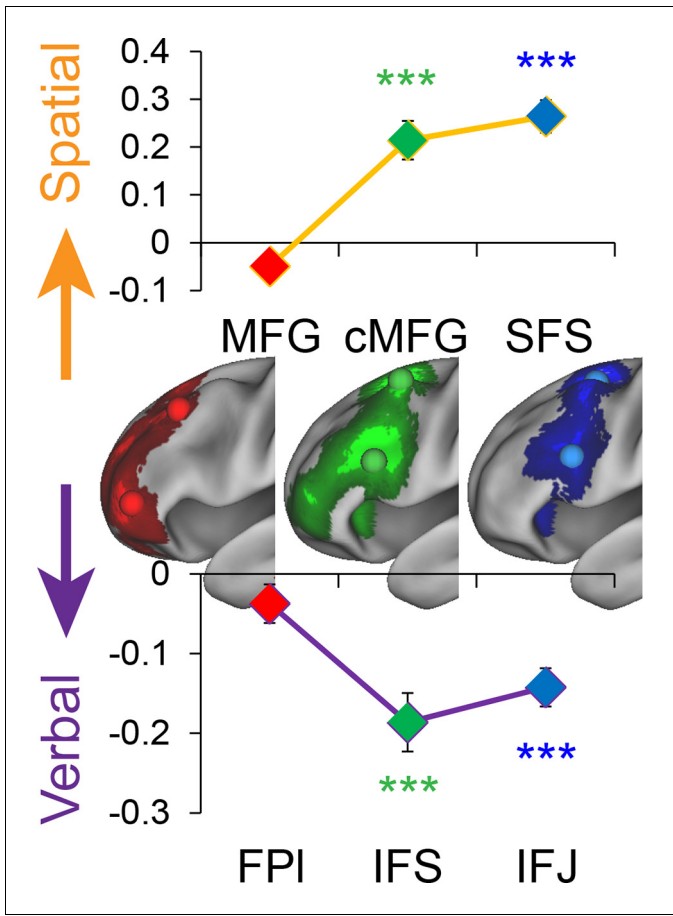

**Figure 4.** Stimulus domain-sensitivity along the rostral/caudal axis. Regions defined by peak effects of *Temporal Control* (red), *Contextual Control* (green), and *Feature Control* (blue). Points reflect the contrast estimates for spatial – verbal such that positive values represent spatial sensitivity and negative values represent verbal sensitivity. Stimulus domain-sensitivity was observed in caudal and mid, but not rostral areas of the LPFC. ***$p_{corrected}<0.0005$.

The following figure supplement is available for figure 4:

**Figure supplement 1.** Stimulus domain-sensitivity along the rostral/caudal axis robustness.

domain, and the close proximity of the peaks of effects of *Stimulus Domain* and *Feature Control* (see Materials and methods; *Figure 3—figure supplement 1*), we use the contrast of *Stimulus Domain* to define caudal areas in future analyses since they enable the most straight-forward separation of spatial and verbal aspects of *Feature Control*.

## Abstraction: temporal control over action

Abstraction along the rostral/caudal axis of the LPFC is hypothesized to relate to the temporal control over action (*Fuster, 1997*; *Koechlin et al., 2003*; *Koechlin and Summerfield, 2007*). By these ideas, concrete processing in caudal LPFC provides control over current processing (e.g. translating stimuli to actions), while abstract processing in rostral LPFC organizes behavior according to future considerations (e.g. goals or plans). To test this proposal, we examined the degree to which different rostral/caudal levels of the LPFC were related to current processing demands and future processing demands, as indexed by RT. Current demands were operationalized as RT during sub-task trials (i.e. RTs corresponding directly to the epochs at which fMRI signal was measured). Future demands were operationalized as RT during return trials (i.e. RTs corresponding to demands the trial after the fMRI signal was measured). Regressing these measures onto activation revealed a clear gradient such that caudal LPFC was related to current, but not future processing, while rostral LPFC was related to future, but not current processing, with mid LPFC showing activity related to both current and future processing (*Figure 5A*). To quantify these relationships, we examined partial correlations with the ROIs described above. This revealed significant relationships between caudal LPFC and current ($r = 0.69$, $p_{corrected} < 10^{-22}$), but not future processing ($r = 0.04$, $p > 0.5$). Conversely, rostral LPFC was related future ($r = 0.31$, $p_{corrected} < 0.0005$), but not current processing ($r = 0.06$, $p > 0.45$). Finally, activity in mid LPFC was related to both current ($r = 0.62$, $p_{corrected} < 10^{-17}$) and future processing ($r = 0.31$, $p_{corrected} < 0.0005$; *Figure 5B*).

To bolster these findings, we regressed current and future RT on activations at each LPFC level in each subject. Then inferences on the resultant slope parameters were performed at the group-level (i.e. summary-statistic approach). These parameters were submitted to a 3 x 2 ANOVA with factors of LPFC level (caudal, mid, rostral) and time (current, future). This analysis revealed a significant LPFC level x time interaction ($F(2,46) = 30.85$, $p < 10^{-8}$; *Figure 5C*). T-tests on the individual parameter estimates corroborated the same pattern (caudal-current: $t(23) = 8.11$, $p_{corrected} < 10^{-6}$; caudal-future: $t(23) = 0.80$, $p > 0.4$; mid-current: $t(23) = 7.21$, $p_{corrected} < 10^{-5}$; mid-future: $t(23) = 3.14$, $p_{corrected} < 0.05$; rostral-current: $t(23) = -0.61$, $p > 0.5$; rostral-future: $t(23) = 3.60$, $p_{corrected} < 0.01$). Collectively, these data indicate a temporal abstraction gradient such that caudal LPFC areas reflect current processing demands, while rostral LPFC areas reflect future processing demands. Mid LPFC areas were related to both current and future demands, suggesting a potential zone for integrating rostral and caudal influences. These effects were replicable when the dataset was split into two, confirming their robustness (*Figure 5—figure supplement 1*).

## Evaluating hierarchical interactions

The above analyses demonstrate that the rostral/caudal LPFC axis is sensitive to abstraction with caudal areas showing concrete properties including stimulus domain-sensitivity and present-oriented processing, and rostral areas showing abstract properties including stimulus domain-generality and future-oriented processing. However, the question remains regarding how these areas interact to support goal-directed cognition and whether such interactions are hierarchical.

Hierarchical interactions are presumed to be reflected by asymmetries in the influence of one region upon another (*Badre and D'Esposito, 2009*). Regions that are high in the hierarchy are thought to influence lower regions more than vice versa. Since directed influences cannot be inferred on the basis of correlations alone, we performed dynamic causal modeling (DCM) which describes directed influences between brain regions (effective connectivity) by estimating parameters in a series of differential equations (*Friston et al., 2003*, see *Stephan et al., 2010* for an accessible review). These differential equations include a forward model linking the hemodynamic response to underlying neural activity in order to describe how activity changes in a region as a function of inputs to that region. The ability of DCM to uncover underlying directed neuronal interactions has been validated using large-scale biophysical models (*Lee et al., 2006*) and invasive recordings in rodents (*David et al., 2008*). However, since computational complexity precludes modeling all sources of

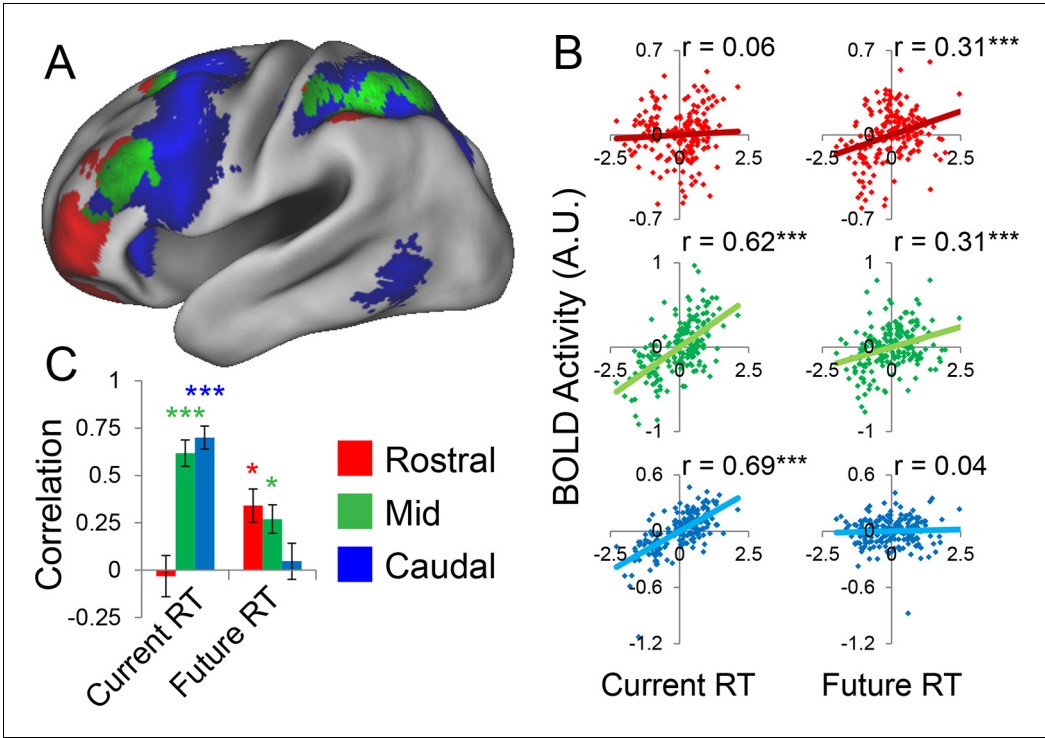

**Figure 5.** Temporal activation-behavior relationships. Correlations between activation and reaction time (RT). Current RT corresponds to sub-task trials while future RT corresponds to return trials. RT measures have been normalized within-subject across the 8 conditions of interest. (A) Voxel-wise regression of Current and Future RT onto activations for the 8 conditions of interest across subjects. Individual subject terms have been regressed out. Red: significant correlations with Future RT only; Blue: significant correlations with Current RT only; Green: both. (B) Partial correlations between activation and RT for the 8 conditions of interest for all subjects. Individual subject terms have been regressed out. Red: rostral LPFC; Green: mid LPFC; Blue: caudal LPFC. (C) Average partial correlation computed separately for each subject (summary-statistic approach). *$p_{corrected}$<0.05; ***$p_{corrected}$<0.0005.

The following source data and figure supplement are available for figure 5:

**Source data 1.** Statistical parametric maps of the activation-behavior correlations.

**Figure supplement 1.** Temporal activation-behavior relationships robustness.

inputs that drive changes in a particular region, DCM can only be applied to simplified neural circuits (i.e. a small number of regions). Such simplifications also require including experimental manipulations such as stimulus delivery and experimental condition as direct drivers and modulators of activity. Despite these simplifications, DCM has been demonstrated to accurately identify the appropriate connections and modulations, even when the modeled regions are only a subset of the regions contributing to a process (*Lee et al., 2006*). Finally, DCM leverages model comparison to determine the presence and strength of directed regional couplings. This requires specifying multiple (plausible) models and comparing model evidence to reveal which model provides the best account of the data. Hence, DCM can only make relative claims with respect to the explored model space.

To examine interactions among LPFC sub-regions, we modeled the dynamics of activity in the six LPFC ROIs as a function of three sets of parameters: 1) inputs into the system from stimuli received by caudal areas; 2) connectivity among regions along which activity is propagated (fixed connectivity); and 3) changes in connectivity induced by demands of stimulus domain and cognitive control (connectivity modulations). An example is schematized in *Figure 6—figure supplement 1*. We then

searched for asymmetries in the fixed connectivity and their modulations for evidence of hierarchical control.

Group-level Bayesian model comparison adjudicated between models of effective connectivity within the LPFC (*Stephan et al., 2009a*) (see Methods for full details of the model procedure). Significant parameter estimates resulting from the best model are depicted in *Figure 6*. We began by examining asymmetries in fixed connectivity (*Figure 6A*). Fixed connectivity is constrained by structural connectivity (*Stephan et al., 2009b*), and thus provides a window into the relative strength of efferent and afferent connections. To evaluate whether fixed connections demonstrate hierarchical dependencies, we calculated the relative strength of efferent versus afferent connections with a simple contrast (average efferent strength – average afferent strength) based on estimated fixed connectivity of the model. Higher hierarchical areas would be expected to show a positive value on this metric. Similar explorations in the monkey have provided evidence for hierarchical structure, but with controversy surrounding whether the rostral LPFC is the apex of the hierarchy (*Badre and D'Esposito, 2009*) or the mid LPFC (*Goulas et al., 2014*). In our dataset, hierarchical strength increased monotonically from caudal areas to mid areas to MFG, but FPl demonstrated the lowest value on this metric resulting in an inverted-U pattern (*Figure 7*). These effects were replicable when the dataset was split into two, confirming their robustness (*Figure 7—figure supplement 1*). This pattern was not due to the average connectivity strength or total connectivity strength, which showed markedly different patterns (*Figure 7—figure supplement 2*), indicating that hierarchical strength is independent of the general magnitude of connectivity.

To better quantify the hierarchical organization of the LPFC, we fit a quadratic (e.g. inverted-U) function to each participants' hierarchical strength metrics. Here, the height of the vertex of the quadratic function estimates the hierarchical strength of the apex of the hierarchy, assuming it is positive. The position of the vertex provides an estimate of the rostral/caudal location of apex of the hierarchy. Across participants, the height of the vertex was significantly positive (t(23) = 2.34, p<0.05), indicating that the vertex tended to represent the apex of the hierarchy. Furthermore, the vertex tended to be localized to the mid LPFC (mean y-coordinate: 24.7, standard error: 6.1) and differed significantly from both the rostral-most (i.e. FPl; t(23) = -4.35, p<0.0005), and caudal-most ROIs (i.e. SFS; t(23) = 4.18, p<0.0005). These results are inconsistent with a strict rostral-to-caudal hierarchy (*Badre and D'Esposito, 2009*) and instead suggest that the apex of the LPFC hierarchy is in mid areas (*Goulas et al., 2014*).

Next, we searched for asymmetries in the modulations of effective connectivity by cognitive demands. Modulatory dynamics (*Figure 6B*) had the following properties: first, *Stimulus Domain* demands increased connectivity from caudal to mid LPFC areas (hereafter, caudal-to-rostral modulations are referred to as 'bottom-up'). That is, when processing spatial stimuli, connectivity was increased from SFS to cMFG (t(23) = 3.70, q<0.005), while processing verbal stimuli increased connectivity from IFJ to IFS (t(23) = 3.89, q<0.001). Second, cognitive control demands modulated connectivity from rostral to mid LPFC areas (hereafter, rostral-to-caudal modulations are referred to as 'top-down'). *Temporal Control* increased connectivity from FPl to MFG (t(23) = 2.40, q<0.05). *Contextual Control* decreased connectivity from FPl to MFG (t(23) = -5.45, q<0.0001), IFS (t(23) = -4.76, q<0.0005), and cMFG (t(23) = -4.15, q<0.001), while increasing connectivity from MFG to IFS (t(23) = 8.18, q<10⁻⁶) and cMFG (t(23) = 7.70, q<10⁻⁶). Collectively, these data indicate that demands on stimulus domain influence bottom-up processing, while demands on cognitive control influence top-down processing. These results suggest that demands on cognitive control result in a convergence of influences at the mid LPFC where top-down and bottom-up information meet. These effects were replicable when the dataset was split into two, confirming their robustness (*Figure 6—figure supplement 2*).

## Relationship between LPFC dynamics and cognitive ability

Hierarchical processing is important due to high degree of flexibility and computational efficiency it confers (*Botvinick and Weinstein, 2014*; *Duncan, 2013*; *Solway et al., 2014*). Therefore, we predicted that the modeled LPFC hierarchical interactions should be related to cognitive abilities. To explore this possibility, we examined the relationship between the modeled estimates of effective connectivity and trait-measured higher-level cognitive ability.

First, we observed that individual differences in the magnitude of top-down modulations by cognitive control (as measured by estimates of modulations of effective connectivity; *Figure 6B*) tended

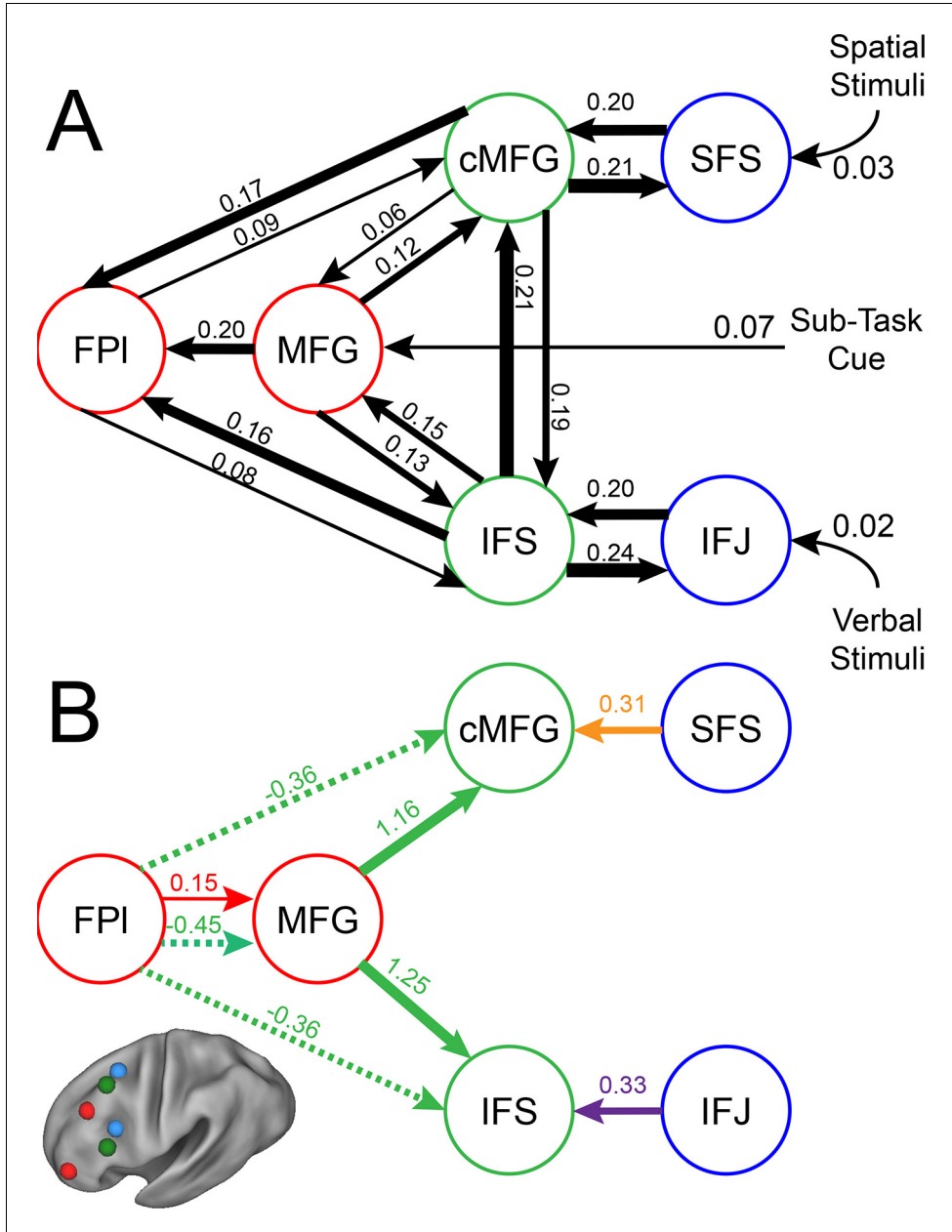

**Figure 6.** LPFC dynamic causal model. Interactions within the LPFC were modeled using dynamic causal modeling. Bayesian model selection indicated that the depicted model was the best model of the dynamics among the models tested. Arrows indicate direction of influence, numbers and line widths indicate the strength of influence, and dashed arrows indicated inhibitory influences. (**A**) Fixed connectivity and inputs depicted in black. (**B**) Modulations of connectivity by *Spatial Stimulus Domain* (orange), *Verbal Stimulus Domain* (purple), *Contextual Control* (green), and *Temporal Control* (red) demands depicted in colors. *Stimulus Domain* demands produced feedforward influences from caudal areas (blue) to mid areas (green). Cognitive control demands produced feedback influences from rostral (red) to mid areas (green). All depicted parameters are significant after correction using false-discovery rate.

The following figure supplements are available for figure 6:

**Figure supplement 1.** Depiction of dynamic causal modeling.

**Figure supplement 2.** LPFC dynamic causal model robustness.

*Figure 6 continued on next page*

*Figure 6 continued*

**Figure supplement 3.** Path coefficients in a rostral-to-caudal model.
**Figure supplement 4.** Connectivity as a function of cost.
**Figure supplement 5.** Model comparison.

---

to be inter-related (*Figure 8—figure supplement 1A*). This suggests that top-down control is an individual difference metric. Using principle components analysis, we distilled the cognitive control-induced changes in top-down effective connectivity into a single factor (*Figure 8—figure supplement 2*). Similarly, we combined the stimulus domain-induced changes in bottom-up effective connectivity into a single factor. We found that these measures were inversely related, such that individuals that tended to have strong top-down influences tended to have weaker bottom-up influences and vice versa (r = −0.52, p<0.01; robust regression t(22) = −2.79, p<0.05; *Figure 8A*). This was not due to the model fitting procedure as the model covariance matrix indicated a negligible relationship between these factors (*Figure 8—figure supplement 1B*).

Next, we examined several independent measures of higher-level cognitive ability which included tests of working memory and fluid intelligence. Performance on these measures tended to be inter-correlated (*Figure 8—figure supplement 1C*), so we combined them into a single factor using principle components analysis. We examined the relationship between model-derived metrics of top-down and bottom-up processing, and trait-measured higher-level cognitive ability. While there was no relationship between bottom-up processing and higher-level cognitive ability (r = −0.01, p>0.95; robust regression t(22) = 0.09, p>0.9; *Figure 8C*), there was a significant positive relationship between top-down processing and higher-level cognitive ability (r = 0.47, p<0.05; robust regression t(22) = 2.25, p<0.05; *Figure 8B*). This relationship remained after removing 2 participants with higher-level cognitive abilities greater than 2 standard deviations below the mean (r = 0.43, p<0.05; robust regression t(20) = 2.21, p<0.05). Hence, those individuals with greater top-down processing demonstrate greater higher-level cognitive ability.

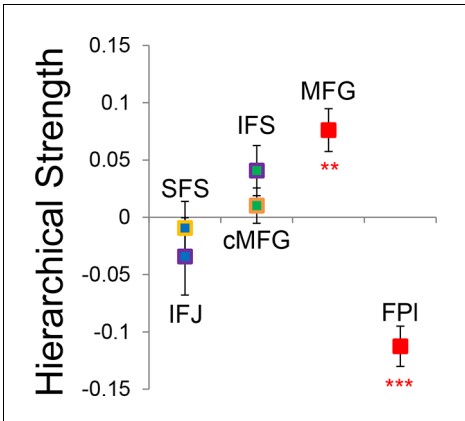

**Figure 7.** Hierarchical structural dependencies. Based on fixed connectivity of the model depicted in *Figure 6A*, hierarchical strength was calculated as the difference between inward and outward projections along the rostral/caudal axis. Hierarchical strength rose monotonically from caudal to mid areas, but fell precipitously at the rostral most portion of the network. **$p_{corrected}$<0.005; ***$p_{corrected}$<0.0005.
The following figure supplements are available for figure 7:

**Figure supplement 1.** Hierarchical structural dependency robustness.

**Figure supplement 2.** Average and total connectivity strength.

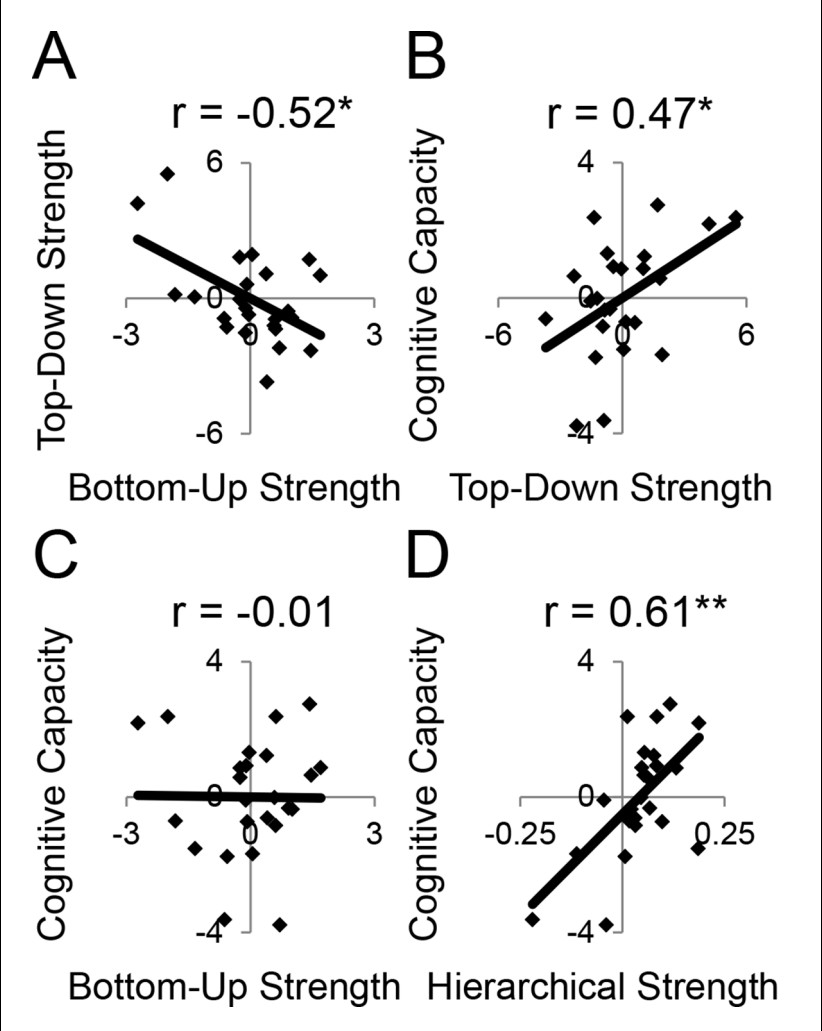

**Figure 8.** LPFC dynamics and higher-level cognitive ability. Neural metrics were based on modeled estimates of effective connectivity and their modulations (*Figure 6*). Metrics reflecting top-down LPFC modulations by cognitive control demands (top-down strength), and metrics reflecting bottom-up LPFC modulations by *Stimulus Domain* demands (bottom-up strength) were combined, respectively. (**A**) Top-down and bottom-up strength were anti-correlated. (**B**) Top-down strength predicted trait-measured higher-level cognitive capacity. (**C**) By contrast, bottom-up strength did not correlate with higher-level cognitive capacity. (**D**) Hierarchical strength reflected the degree to which mid LPFC showed greater outward than inward fixed connectivity. This metric was also positively related to higher-level cognitive capacity. *p<0.05; **p<0.005.

The following figure supplements are available for figure 8:

**Figure supplement 1.** Model and trait correlations and covariances.

**Figure supplement 2.** Dynamic causal modeling derived individual difference measures.

**Figure supplement 3.** LPFC dynamics and higher-level cognitive ability robustness.

Subsequently, we explored the relationship between the hierarchical strength of the modeled fixed connectivity and trait-measured higher-level cognitive ability. In particular, we were interested in whether the degree to which mid LPFC demonstrated hierarchically organized fixed connectivity predicted higher-level cognitive ability. We related the height of the vertex fit described above – summarizing the hierarchical strength of mid LPFC – to trait-measured higher-level cognitive ability.

This relationship was significantly positive (r = 0.61, p<0.005; robust regression t(22) = 4.68, p=0.0001; *Figure 8D*) indicating that those individuals with stronger mid LPFC hierarchical fixed connectivity demonstrate greater higher-level cognitive ability. This relationship largely remained after removing 2 participants with higher-level cognitive abilities greater than 2 standard deviations below the mean (r = 0.39, p=0.07; robust regression t(20) = 3.00, p<0.01). Furthermore, this relationship remained after controlling for contributions of average connectivity of mid LPFC (cMFG, IFS, and MFG), and global connectivity of the LPFC (ρ = 0.47, p<0.05; robust regression t(20) = 2.66, p<0.05), indicating that hierarchical connectivity makes important contributions to higher-level cognitive ability over-and-above simple connectivity magnitude. This suggests that the hierarchical strength of the mid LPFC is beneficial to higher-level cognition.

The height of the vertex summarizes the hierarchical strength of mid LPFC. However, it is obtained by fitting the hierarchical strengths of all areas, thereby incorporating some information from other regions, as well. One might be interested in the degree to which the relationship between higher-level cognitive ability and hierarchical strength is selective to mid LPFC. This is difficult to disentangle due to the fact that hierarchical strengths of different regions are interdependent. Nevertheless, a simple average of the mid LPFC hierarchical strength (including cMFG, IFS, and MFG) produced similar results to the height of the vertex method described above (r = 0.48, p<0.05; robust regression t(22) = 2.38, p<0.05). Opposite results were obtained by relating higher-level cognitive ability to the hierarchical strength of caudal LPFC (r = −0.43, p<0.05; robust regression t(22) = −2.00, p=0.057) and FPl (r = −0.42, p<0.05; robust regression t(22) = −2.41, p<0.05), which is to be expected given the strong interdependencies between the hierarchical strength of mid LPFC to both caudal and rostral LPFC (mid to caudal, r = −0.84, p<10$^{-6}$; mid to FPl, r = −0.87, p<10$^{-7}$). Partial correlations were unable to convincingly determine whether the relationships between higher-level cognitive ability and hierarchical strength were primarily due to the hierarchical strength of mid LPFC or other areas ($\rho_{mid \cdot caudal}$ = 0.27, p>0.2; $\rho_{mid \cdot FPl}$ = 0.29, p>0.15; $\rho_{caudal \cdot mid}$ = −0.03, p>0.85; $\rho_{FPl \cdot mid}$ = 0.03, p>0.85). However, that the relationships between caudal/rostral LPFC and higher-level cognitive ability were negligible after partialing out the contributions of mid LPFC and that the reverse was not true is suggestive that the main driver is the hierarchical strength of the mid LPFC.

Finally, we assessed the unique contribution of each LPFC dynamic factor described above to higher-level cognitive ability. The strength of top-down modulations by cognitive control, bottom-up modulations by stimulus domain, and hierarchical fixed connectivity of the mid LPFC were simultaneously regressed on trait-measured higher-level cognitive ability. This analysis revealed significant effects of top-down modulations (ordinary least-squares regression t(20) = 2.45, p<0.05; robust regression t(20) = 2.21, p<0.05), and mid LPFC hierarchical fixed connectivity (ordinary least-squares regression t(20) = 3.13, p<0.01; robust regression t(20) = 3.58, p<0.005), but no effect of bottom-up modulations (ordinary least-squares regression t(20) = 1.81, p>0.05; robust regression t(20) = 1.48, p>0.15). These results indicate that both the hierarchical orientation of the mid LPFC at baseline, and the top-down modulation from rostral LPFC to mid LPFC during cognitive control make separable and positive contributions to higher-level cognitive ability.

## Discussion

We examined interactions within the LPFC by varying demands on stimulus domain and cognitive control. Collectively, these demands engaged most of LPFC providing the opportunity for a comprehensive account of its function. We found that progressively rostral areas processed progressively abstract levels of cognitive control with demands on *Feature Control, Contextual Control*, and *Temporal Control* eliciting increasingly rostral activations. Moreover, stimulus domain-sensitivity was present in caudal and mid LPFC, but not in rostral LPFC consistent with abstract processing in rostral areas. Correlations between neural activation and behavioral performance revealed that caudal LPFC varied linearly with demands on current processing, while rostral LPFC varied linearly with demands on future processing. Both relationships were present in the mid LPFC, indicating a potential zone where current and future processing converges. Dynamic causal modeling revealed that hierarchical asymmetries in connectivity were strongest in mid LPFC. Furthermore, cognitive demands elicited both top-down influences from rostral to mid LPFC and bottom-up influences from caudal to mid LPFC. Thus, analyses of effective connectivity and its modulations suggest that mid

LPFC forms a nexus where information converges to influence action. Finally, both the hierarchical orientation of the mid LPFC in fixed connectivity, and the top-down influences from rostral LPFC to mid LPFC during the engagement of cognitive control were positively related to higher-level cognitive ability. Hence, the functional organization of the LPFC is intimately tied to cognitive capacity.

A matter of debate is whether or not processing in the LPFC is hierarchical. From a computational perspective, hierarchical processing is advantageous (*Botvinick and Weinstein, 2014*; *Solway et al., 2014*), although relating such processing advantages to the LPFC has been challenging. Badre and D'Esposito (*Badre and D'Esposito, 2009*) suggested that the hallmark of hierarchies is asymmetrical influence. That is, those components that are higher in the hierarchy influence lower components more than vice versa. Based on patterns of efferent and afferent projections, they proposed a rostral-to-caudal hierarchy such that rostral areas are higher than caudal areas. Recently, however, the evidence for a rostral-to-caudal hierarchy based on asymmetry of anatomical projections has been called into question. Based on a meta-analysis of projections among areas of the LPFC in monkeys, hierarchy was determined by the asymmetry with which a given area sent projections to other areas compared to the converse (*Goulas et al., 2014*). The results did not support the notion that rostral LPFC (frontopolar cortex, putative area 10) is the top of the hierarchy. Instead, mid LPFC areas (putative areas 45 and 46) tended to be at the highest hierarchical levels, while area 10 was low in the hierarchy. However, homologies between human and monkey LPFC are still uncertain and there are marked differences between human and monkey frontopolar cortex (*Semendeferi et al., 2001*; *Neubert et al., 2014*). Nevertheless, our results examining fixed connectivity, a putative correlate of structural connections, are consistent with the findings that mid LPFC is higher and rostral LPFC is lower in the LPFC hierarchy. Collectively, these data suggest that mid LPFC forms the apex of the frontal hierarchy as measured by asymmetrical connectivity.

Other support for a rostral-to-caudal LPFC hierarchy comes from observed changes in functional connectivity by cognitive demands. In a landmark study, Koechlin et al (*Koechlin et al., 2003*) demonstrated a cascade of activations across LPFC such that the more temporally remote the control process, the more rostral the activation. Moreover, structural equation modeling revealed significant effective connectivity from rostral to caudal areas as a function of cognitive demands, consistent with a rostral-to-caudal hierarchy. While compelling, rostral-to-caudal models were not compared to other organizational schemes. Since a different organization may have produced superior fits to the data, it is difficult to draw conclusions from these connectivity data. In the present dataset, a strictly rostral-to-caudal model of modulations produced significant top-down modulatory effects consistent with Koechlin et al (*Koechlin et al., 2003*). However, this model produced a substantially worse fit to the data than the model we describe (see *Figure 6—figure supplement 3*), rendering rostral-to-caudal dynamics doubtful.

In the present data, we performed explicit model comparison and examined effective connectivity as the LPFC was engaged by a variety of cognitive demands. We found that influences as a function of cognitive demands did not follow a strict rostral-to-caudal progression. While cognitive control demands did modulate influences from rostral to mid areas, stimulus domain demands modulated influences from caudal to mid areas. This pattern of connectivity is consistent with the notion that mid areas are an integration zone where domain-general top-down influences and domain-specific bottom-up influences converge. This is corroborated by the activation-behavior relationships which demonstrated that whereas rostral and caudal areas were related to future and current processing, respectively, mid areas were related to both. This suggests that mid LPFC simultaneously represents information from both rostral and caudal areas. From a developmental perspective, it is interesting to note that mid LPFC areas develop later than caudal and rostral LPFC areas (*Shaw et al., 2008*; *Gogtay et al., 2004*). As a general pattern, integrative areas tend to develop later than the areas over which they integrate (*Gogtay et al., 2004*). Thus, the maturation of mid LPFC areas may be influenced by simultaneous top-down and bottom-up demands (*Badre and D'Esposito, 2009*) preparing the mid LPFC for an integrative role. These ideas are bolstered by evidence that the mid LPFC is a cortical hub with high global connectivity, the degree of which predicts intelligence (*Cole et al., 2012*). If mid LPFC is a convergence zone where top-down influences from rostral LPFC and bottom-up influences from caudal LPFC meet, it could serve to integrate abstract goals or intentions from rostral areas and concrete contextual information from caudal areas in order to guide action selection.

Perhaps the strongest data to date for rostral/caudal LPFC asymmetries comes from lesion studies. Badre et al (*Badre et al., 2009*) manipulated the level of cognitive control while testing patients with focal lesions to different LPFC regions. They found that those patients that had damage to mid LPFC demonstrated impaired performance at higher levels of the task, but not lower levels, while patients that had damage to a caudal LPFC demonstrated impaired performance on both higher and lower levels of the task. These results were recently replicated and extended with a different task whereby similar asymmetries were observed across three levels: caudal, mid, and rostral LPFC (*Azuar et al., 2014*). Collectively, these studies suggest a rostral/caudal hierarchy wherein higher level controllers depend upon intact lower level controllers, but not vice versa.

Both the model of the present data (*Figure 6*) and the lesion studies are consistent with the proposal that operations of the mid LPFC depend upon caudal LPFC, but not vice versa. As depicted in the model, this is due to bottom-up, but not top-down, modulations from caudal LPFC to mid LPFC as a function of cognitive demand. Of note is that mid LPFC lesions in the studies of Badre et al (*Badre et al., 2009*) and Azuar et al (*Azuar et al., 2014*) appeared to include the area we refer to as MFG (putative area 9/46). In our model, MFG processes sub-task cues, consistent with its role in rule processing (*Nee and Brown, 2012*; *Bunge et al., 2005*; *Genovesio et al., 2005*; *Wallis et al., 2001*). Given that all of the cognitive control demands require processing the sub-task cues, lesions to MFG would affect all of the cognitive control sub-tasks. In this way, operations of FPl are dependent on an intact MFG. Furthermore, in the model, modulations from FPl onto mid LPFC are inhibitory in nature, counteracting the baseline positive influence observed in the fixed connectivity. Thus, the cancellation of negative and positive influences could lead to a negligible impact of FPl lesions on mid LPFC processing. Collectively then, our model indicates that a similar rostral/caudal cascade of asymmetric impairments could be observed with the present task dynamics. Furthermore, our data highlight the need for task-based modeling of neural dynamics in the intact brain to unravel the mechanisms that can lead to lesion-based deficits.

The parameters estimated in our model indicated both positive (i.e. excitatory) and negative (i.e. inhibitory) modulations of effective connectivity. Interestingly, all inhibitory modulations arose from FPl as a function of *Contextual Control* demands. That is, *Contextual Control* reduced the influence of this region upon more caudal areas. This is in contrast to the excitatory modulation from FPl to MFG observed during *Temporal Control* demands. While *Temporal Control* required preserving past information in order to integrate it into future processing, *Contextual Control* required the isolation of current processing from previous and future processing. Our data suggest that the dynamics of FPl reflect these two forms of processing. That is, excitatory influence from FPl promotes integrative processing, while inhibitory influence from FPl promotes segregated processing. Such inhibitory influences during *Contextual Control* may have been necessary to overcome the positive fixed connectivity observed between FPl and other regions. Hence, although FPl was active for both *Temporal Control* and *Contextual Control*, this activity promoted different forms of processing under different conditions, realized by dynamic interactions between FPl and other areas of the LPFC.

Our data highlight that directions of influence are dynamic rather than static. That is, whereas fixed connectivity demonstrated asymmetrically greater influence from mid LPFC to other regions, various task demands modulated connectivity towards, but not from, the mid LPFC. This suggests that asymmetries in the LPFC shift according to task demands. Recent work has demonstrated that the LPFC flexibly adapts its connectivity as a function of task demands (*Cole et al., 2013*), suggesting that dynamics in the LPFC may be of central importance for cognitive control. Whether and how such dynamics support hierarchical processing is a critical matter for future research.

Connectivity modulations that did follow a rostral-to-caudal direction of influence positively predicted independent measures of higher-level cognitive ability. However, modulations in the opposite direction demonstrated no such relationship. While it has been well-documented that top-down influences from the LPFC to posterior areas of cortex are important for working memory, sustained attention, mitigating distraction, and a variety of other cognitive abilities (*D'Esposito and Postle, 2015*; *Gazzaley and Nobre, 2012*), the relationship between top-down control *within* the LPFC and such higher-level cognitive processes has been less well studied. Our data suggest that top-down control within the LPFC is an important determinant of higher-level cognitive function. Furthermore, we found that more strongly asymmetrical hierarchical projections from mid LPFC to other LPFC

regions were related to better higher-level cognitive ability. This suggests a beneficial cognitive effect of a hierarchical organization with mid LPFC at the apex.

The term 'hierarchy' is typically used to describe dominance relationships. Accordingly, we have attempted to operationalize such dominance relationships by reasoning that greater outward relative to inward connectivity reflects larger influence of one region over another. Such rationale has been used previously (*Badre and D'Esposito, 2009*; *Goulas et al., 2014*). However, different definitions of hierarchy may lead to different assignments of rank among the LPFC. For example, hierarchy may be defined by temporal control, with higher levels organizing behavior over broader timescales (*Koechlin et al., 1999*; *2003*). Alternatively, hierarchy could be defined by modulations of influence as a result of cognitive demands. Such relationships are typified by the top-down biasing that LPFC exerts over posterior areas during difficult selections (*Miller and Cohen, 2001*). These various definitions lead to different rank orderings in our data. However, regardless of definition, the mid LPFC appeared to play a central role in our data showing the greatest outward relative to inward asymmetry, containing both future and present oriented signals that related to behavior, and providing a convergence zone of modulatory inputs. Hence, these data make it clear that the mid LPFC is a critical nexus for cognitive control.

How to operationally define the rostral/caudal axis of the LPFC has been a matter of considerable debate. Gradients of activation along the rostral/caudal axis of the LPFC have been observed by manipulating the temporal timescale of control processing and relational complexity (see *Badre, 2008* for an in-depth review). However, activations spanning the rostral/caudal axis of the LPFC can be observed by tasks that do not clearly manipulate these processes (*Crittenden and Duncan, 2014*), and manipulations of these processes do not always lead to orderly gradients of activation (*Reynolds et al., 2012*). The purpose of the present study was not to resolve these issues (but see *Nee et al., 2014* for a recent attempt). However, our results are most consistent with the idea that the LPFC is temporally organized (*Koechlin et al., 2003*; *Koechlin and Summerfield, 2007*; *Fuster, 2001*). In these data, caudal LPFC was involved in current processing, providing selective attention to visual stimulus features, while rostral LPFC was involved in future processing, enabling the retention of information for integration into future processing. The mid LPFC appeared to synthesize both current and future processing allowing the use of current and future informed contextual information to organize behavior. Hence, the temporal timescale of processing/representation increased from caudal to rostral areas.

A rostral-to-caudal hierarchy makes intuitive sense from the perspective of the temporal control of action. Long-lasting abstract representations such as goals or plans are thought to influence concrete representations such as actions. Mid-level representations such as contexts are in-between, governing the correct action tendencies given the prevailing situation. By a rostral-to-caudal hierarchy then, goals should govern contexts. However, it is often the case that contexts dictate whether a goal is achievable or not (e.g. a social situation is not a good context within which to write a manuscript). In such cases, the context may suggest a new goal to pursue (e.g. network instead of write). Hence, the ideal controller may take into account longer-term objectives and the current situation in order to arrive at the correct behavior for the moment. Our data indicate that the mid LPFC has the connectivity properties of such a controller, thereby forming a central nexus for cognitive control.

## Materials and methods

### Participants

We report results from 24 (13 female) right-handed native English speakers (mean age 19.9 years, range 18–28). The targeted number of participants was based upon previous work indicating that sample sizes in the 20–25 range are appropriate to achieve reliable fMRI results (*Thirion et al., 2007*) and within the range of previous relevant literature (*Badre and D'Esposito, 2007*; *Nee and Brown, 2012*; *2013*; *Bahlmann et al., 2015*; *Nee et al., 2014*; *Jeon and Friederici, 2013*). Informed consent was obtained in accordance with the Committee for Protection of Human Subjects at the University of California, Berkeley. Data from only the first session are reported for one participant since that they were unable to complete the second fMRI session due to technical issues.

## Procedure

The task design was a factorial 2 x 2 x 2, with factors of *Stimulus Domain* (verbal, spatial), *Contextual Control* (high, low), and *Temporal Control* (high, low; *Figure 1*). 'Contextual Control' in the present paradigm was originally referred to as 'dual-tasking' (*Koechlin et al., 1999*). Later (*Charron and Koechlin, 2010*), it was recognized that the dual-tasking component applies most aptly to the condition we refer to as 'Dual' (also referred to as 'Branching' [*Koechlin et al., 1999*; *Charron and Koechlin, 2010*]). 'Switching' was the term applied in more recent work (*Charron and Koechlin, 2010*). However, the examined process is likely to be akin to preparing to switch rather than switching itself. The process by which a contextual cue determines the appropriate rule set has previously been referred to as 'Contextual Control' (*Koechlin et al., 2003*; *Koechlin and Summerfield, 2007*), which we feel accurately describes the process. 'Temporal control' in the present paradigm was originally referred to as 'working memory' (*Koechlin et al., 1999*). Di Pisapia et al (*De Pisapia et al., 2007*) have demonstrated that relative to simple maintenance, to which working memory is often associated, maintenance for future integration specifically engages rostral LPFC. It is this process that is of primary interest here. In Koechlin's framework (*Koechlin et al., 2003*; *Koechlin and Summerfield, 2007*), the process has partial overlap with both 'episodic control' and 'branching control', both of which are distinguished from other forms of control by their temporal nature. Here, we use the term 'temporal control' which we believe most generally reflects the process.

On each trial, a letter was presented at one of five spatial locations (i.e. points of a star) surrounded by a colored frame. The color informed participants whether letters or locations were relevant for the block of 7 to 13 trials. Color-to-stimulus domain mappings were counterbalanced across participants. In the basic tasks, participants determined if the current stimulus followed the preceding stimulus in a sequence (*Figure 1C*). In the first trial of any block, participants decided whether the stimulus was the first item in the sequence (i.e. 't' if the verbal task, the top point of the star if the spatial task). Thereafter, responses depended upon whether the current stimulus followed the preceding stimulus in the sequence. Participants responded either affirmatively or negatively on each trial via keypress using their index fingers with response mappings counter-balanced across participants.

Square frames cued the basic task and all trials of the *Control* condition. Triangle, cross, and circle frames indicated sub-tasks with cue-to-sub-task mappings counterbalanced across participants. All blocks began with 2 to 5 trials of the basic task. Then, either the basic task continued (*Control*), or one of three sub-tasks were denoted by a cue change. In the *Restart* condition, participants determined whether the stimulus was the start of the sequence, thus starting the sequence anew. When the cue reverted to a square, participants once again started a new sequence. In the *Delay* condition, participants ignored all cued stimuli by responding negatively, and resumed the basic task when the cue reverted, evaluating whether the current stimulus sequentially followed the stimulus prior to the cue change. Finally, the *Dual* condition combined both the *Restart* and *Delay* conditions. Participants were cued to begin a new sequence, while also retaining in memory the item prior to the cue change. Each sub-task lasted 3 to 5 trials. Following each sub-task, participants performed a return trial, then 1 to 2 trials of the basic task.

Participants completed 24 blocks of each cell of the 2 x 2 x 2 design over the course of 2 fMRI sessions. A total of 1920 trials were performed, including 96 for each sub-task. Sessions were split into 6 runs of 16 blocks (160 trials) each. The experiment length was chosen to match previous research with a related design (*Charron and Koechlin, 2010*). Within a week prior to scanning, participants performed a practice session to learn the task. During the practice session, participants performed the task under experimenter supervision until they were comfortable with the instructions. Thereafter, they completed 3 runs of the task on their own. Furthermore, participants performed a single practice run in the scanner during each session, just prior to functional data collection.

## Measures of higher-level cognitive ability

To obtain independent assessments of higher-level cognitive ability, we collected simple and complex measures of verbal and spatial working memory, as well as fluid intelligence. Complex span was tested using automated measures of verbal working memory (operation span) and spatial working memory (symmetry span), which have been previously described (*Turner and Engle, 1989*). Tests of simple working memory included letter span (verbal) and spatial span (spatial). These tests were also

automated and made identical to the memory portions of the operation span and symmetry span tasks, respectively. Fluid intelligence was measured using Raven's Advanced Progressive Matrices. These measures were collected for the purpose of providing correlates for different aspects LPFC function. However, given the high correlations between these measures (*Figure 8—figure supplement 1C*), the measures were difficult to factorize into dissociable components. As a result, the measures were combined into a single factor reflecting general higher-level cognitive ability using principle components analysis.

## Image acquisition

Images were acquired on a Siemens TIM/Trio 3T MRI equipped with a 32-channel head coil located in the Henry H. Wheeler, Jr. Brain Imaging Center at the University of California, Berkeley. Stimuli were presented to the participant via a coil attached mirror reflecting a projector situated at the bore of the magnet. Experimental tasks were presented using E-Prime software version 2.0 (Psychology Software Tools, Inc., Pittsburgh, PA). Eye tracking was performed using an Avotec system (http://www.avotecinc.com/) and Viewpoint software (http://www.arringtonresearch.com/). Eye position was also monitored by the experimenter during the course of the session for saccades and vigilance, largely to confirm that participants did not use gaze location to assist spatial memory. Since experimenter monitoring confirmed that participants maintained central fixation and vigilance, eye movement data were not further analyzed. Response data were collected on an MR-compatible button box.

Functional T2*-weighted images were acquired using an EPI sequence with 35 descending slices and 3.44 x 3.44 x 3.75 mm$^3$ voxels (TR = 2000 ms; echo time = 25 ms; flip angle = 70; field of view = 220). Three dummy acquisitions preceded each functional scan to allow for image stabilization. Phase and magnitude images were collected to estimate the magnetic inhomogeneity. High-resolution T1-weighted MPRAGE images were collected for spatial normalization (240 x 256 x 160 matrix of 1 mm$^3$ isotropic voxels; TR = 2300 ms; echo time = 2.98 ms; flip angle = 9). A 6-min eyes open resting state scan was collected in each session in addition to the task scans. The resting state scan was collected prior to the task in the first session and after the task in the second session. Imaging data can be made available upon request.

## Image preprocessing

Unless otherwise specified, preprocessing was performed using SPM8 (http://www.fil.ion.ucl.ac.uk/spm/). Images were converted from DICOM into nifti format. Origins for all images were manually set to the anterior commissure. Functional data were spike-corrected to reduce the impact of artifacts using AFNI's 3dDespike routine (http://afni.nimh.nih.gov/afni). Functional images were corrected for differences in slice timing using sinc-interpolation and head movement using a least-squares approach and a 6 parameter rigid body spatial transformation. Images were corrected for distortion and movement-by-susceptibility artifacts using the FieldMap toolbox (*Andersson et al., 2001*). Structural data were coregistered to the functional data and segmented into gray and white-matter probability maps (*Ashburner and Friston, 1997*). These segmented images were used to calculate spatial normalization parameters to the MNI template, which were subsequently applied to the functional data. As part of spatial normalization, the data were resampled to 2 x 2 x 2 mm$^3$. 8-mm full-width/half-maximum isotropic Gaussian smoothing was applied to all functional images. All analyses included a temporal high-pass filter (128 s), correction for temporal autocorrelation using an autoregressive AR(1) model, and each image was run-wise scaled to have a global mean intensity of 100.

## Univariate image analysis

Analyses were performed using SPM8. Subject-level models were fit with a general linear model including epoch regressors for the *Control, Restart, Delay*, and *Dual* conditions crossed with two *Stimulus Domain* types. These regressors spanned the onset of the second trial of the sub-task through the last trial of the sub-task. Since the *Control* condition did not contain an overt sub-task, epochs in the middle of the block were modeled to match the other conditions. Separate impulse regressors were included for the first trial of each block, the first trial of each sub-task, return trials, pre-sub-task trials, and post-sub-task trials, all crossed with *Stimulus Domain*. Additional nuisance

impulse regressors separately coded for left and right motor responses, and error responses. All of the above regressors were convolved with a canonical hemodynamic response function. For participants demonstrating greater than 3mm/degrees of motion over the course of the session or a single movement of greater than 0.5mm/degrees in-between TRs, 24 motion regressors were included reflecting total displacement, squared total displacement, differential (TR-to-TR) displacement, and squared differential displacement to capture signal artifacts related to motion (*Lund et al., 2005*; *Satterthwaite et al., 2013*).

Eight parameters of interest (*Control, Restart, Delay, Dual* conditions each with two *Stimulus Domain* types) were carried forward to a group-level model. This model was estimated as a 2 x 2 x 2 ANOVA with factors of *Stimulus Domain* (verbal, spatial), *Contextual Control* (high, low), and *Temporal Control* (high, low). All main effects and interactions were conducted as t-tests within the ANOVA framework. Main effects of *Stimulus Domain* were explored in both directions (i.e. verbal > spatial, and spatial > verbal). A p<0.001 height and 124 voxel extent threshold provided family-wise error correction at p<0.05 according to the AlphaSim routine in AFNI.

## Robustness and ROI analysis

To determine the robustness of the whole-brain results, we split the dataset in two using an alternating runs approach. For each participant, odd runs were placed in one dataset and even runs in another, counterbalanced across participants. One participant who completed only one fMRI session was excluded from this analysis. Whole-brain analyses were estimated for each dataset as described above. These results can be visualized in *Figure 3—figure supplement 5*.

To provide unbiased estimates for statistical tests and visualization, regions-of-interest (ROIs) were defined in one dataset and tested in the other. To test stimulus domain-sensitivity of regions involved in cognitive control along the rostral/caudal axis, ROIs were placed at activation peaks in the LPFC defined by the main effects of *Temporal Control, Contextual Control*, and their interaction relating to *Feature Control*. Areas of interest were first determined based on the entire dataset, then analogous peaks were localized in each dataset half. Distinct peaks were defined as those at least 1.5 cm from another peak, starting with the *Temporal Control* contrast, followed by the *Feature Control* contrast, followed by the *Contextual Control* contrast, proceeding from the most significant to least significant peak. Different orderings of these definitions produced similar results. Given that the preponderance of activations were left lateralized, consistent with previous reports (*Badre and D'Esposito, 2007*; *Nee and Brown, 2012*; *2013*; *Bahlmann et al., 2014*; *2015*; *Nee et al., 2014*; *Jeon and Friederici, 2013*; *Jeon et al., 2014*), we focused on the left hemisphere. With these procedures, the following areas were defined: FPl (-44 48 4 – whole; -32 58 12 – set 1; -34 54 8 – set 2), rMFG (-28 56 22 whole; -28 56 22 set 1; -28 48 26 set 2), MFG (-38 28 44 – whole; -38 28 44 – set 1; -38 28 46 – set 2), IFS (-52 20 28 – whole; -42 34 16 – set 1; -44 20 28 – set 2), cMFG (-34 10 60 – whole; -30 12 60 – set 1; -24 6 52 – set 2), IFJ (-40 10 20 – whole; -38 10 24 – set 1; -36 6 34 – set 2), SFS (-24 4 54 – whole; -24 4 52 – set 1; -24 4 50 – set 2). For IFS, a right, but not left hemisphere peak was identified in set 1, so a corresponding left hemisphere peak was defined by flipping the sign of the x-coordinate. A 6 mm sphere was created around each peak and spatially-averaged parameter estimates were extracted from each ROI. Due to its lack of connectivity with the rest of the LPFC (described in more detail below; *Figure 6—figure supplement 4*), we describe results in rMFG in the text, but exclude it from the main visualizations and analyses for the sake of expedience. We note that its inclusion does not alter the results or conclusions.

## Relationships with behavior

Relationships between activation and RT were analyzed to assess the role of different areas of the LPFC in cognitive performance. Mean RT for the 8 conditions of interest were calculated for each subject, then z-scored within-subject to produce normalized measures (note that the fMRI data are also scaled prior to parameter estimation). This procedure was done separately for the sub-task trials and the return trials. We refer to RT from sub-task trials as 'Current RT' since these RTs correspond to the epoch at which the fMRI contrasts are measured. We refer to RT from return trials as 'Future RT' since these RTs correspond to the trial following the epoch at which the fMRI contrasts are measured.

Relationships between activation and behavior were assessed in four ways. First, Current and Future RT were regressed onto activation on a voxel-wise basis after regressing out subject-specific effects using a dummy variable approach. Significance was assessed for each relationship with a p<0.001 height and 124 voxel extent threshold providing family-wise error correction at p<0.05 according to AlphaSim. Voxels were then labeled as those showing a significant relationship to Current, but not Future RT, Future, but not Current RT, and both. Subsequent tests were performed within ROIs. We defined rostral LPFC by pooling activations in FPl and MFG, mid LPFC by pooling IFS and cMFG, and caudal LPFC by pooling IFJ and SFS. Areas were defined based upon peaks of the entire dataset described above with the exception that caudal areas were defined based on the main effect of *Stimulus Domain*, which produced ROIs slightly more distanced from the mid LPFC (SFS: -22 0 54; IFJ: -38 6 26). These caudal ROIs were used for all subsequent analyses. Second, partial correlations between activation and Current RT (controlling for Future RT), and activation and Future RT (controlling for Current RT) were calculated after regressing out subject-specific effects using a dummy variable approach. The above two analyses were used for visualization since they produce the most straightforward means to depict the observed relationships. However, to better statistically control for subject-specific effects, we used a summary statistic approach by regressing Current and Future RTs on activations separately for each LPFC zone in each subject and then performing statistics on the corresponding regression slopes. We also performed a linear mixed effects approach that is not described due to redundancy. All four approaches yielded consistent results. All ROI-based tests were Bonferroni corrected for multiple comparisons.

## Resting state reprocessing

Preprocessing was identical to the task-based data described above through co-registration. Linear, squared, differential, and squared differential motion parameters were used as nuisance regressors (*Satterthwaite et al., 2013*). The functional data and nuisance regressors were high-pass filtered at 0.009 Hz. Thereafter, signal from white matter and ventricles were extracted and added as additional nuisance regressors. After regressing out the nuisance regressors, the data were low pass filtered at 0.08 Hz. Finally, the data were spatially normalized and smoothed with an 8-mm Gaussian kernel.

## Dynamic causal modeling

To examine interactions among LPFC sub-regions as a function of cognitive demands, we used dynamic causal modeling (DCM) (*Friston et al., 2003*). DCM is a model-based method for inferring the directed influences among regions, and how these influences are modulated by cognitive demands see (*Stephan et al., 2010*) for an accessible review. DCM consists of two main components: a hemodynamic forward model which describes the transformation of synaptic activity to the hemodynamic response, and a bilinear model that describes how activity changes as a function of inputs, connections, and modulations. For expediency, we focus of the latter. Formally, the bilinear model consists of the following equation:

$$\frac{dx}{dt} = [A + \sum_{j=1}^{m} u_j B^{(j)}]x + Cu$$

Here, *x* denotes the state of the system. In other words, *x* describes the activity level of each region in the modeled system. *u* denotes inputs into the system, which are the external inputs applied to the system (i.e. stimuli). Given *x* and *u*, DCM estimates the dynamics of the system (i.e. $\frac{dx}{dt}$) by estimating three parameters: 1) *A*, which describes the fixed connectivity between regions in the system, 2) *C*, which describes the sensitivity of different regions to external inputs, and 3) *B*, which describes the modulations of connectivity as a function of experimental manipulation *j*.

Our model included FPl, MFG, IFS, cMFG, IFJ, and SFS described above. ROIs were identified in each individual through main effects contrasts of *Temporal Control* (FPl and MFG), *Contextual Control* (IFS and cMFG), and *Stimulus Domain* (IFJ and SFS). 6 mm spherical ROIs were centered around each individual's activation peak closest to the peak in the respective group contrast. In most cases (121/144, 84%), a peak could be identified at whole-brain corrected levels. In the few cases (7/144, 5%) where a peak could not be identified at a lenient p<0.05 uncorrected threshold, the peak from the respective group contrast was used.

Fixed connections were based upon documented structural connectivity in the LPFC (reviewed in *Badre and D'Esposito, 2009*). All fixed connections were assumed to be bidirectional with the strength of connectivity along each direction free to vary. Our fixed connection assumptions were further tested by examining interconnectivity among our LPFC ROIs in resting state data collected during each fMRI session. For each ROI, we calculated connectivity for all other voxels of the brain through a simple time-series correlation. Next, we sorted these connectivity values, excluding voxels within 1.5 cm of the center of the ROI. To determine whether an LPFC region, *r*, was connected to the ROI, we asked whether connectivity between the ROI and *r* fell within the top *c*% of connectivity values, where *c* can be considered the cost. Connectivity was assessed at a range of costs from 5 to 35. At low costs, connectivity was sparse and at high costs connectivity was dense (*Figure 6—figure supplement 4*). Therefore, we chose a cost in the middle of our range (18%) that was consistent with other studies in our laboratory (*Gratton et al., 2012*), and which produced a connectivity structure that resembled documented structural connectivity. Notably, we initially included rMFG in these analyses and found that rMFG was only connected to other LPFC regions at the very peak of our cost range (>34). Thus, it was deemed that rMFG was part of a distinct network and it was not included in DCM and other analyses.

In all models, we assumed that letter/location inputs arrived at the caudal-most aspects of the system, consistent with the dense connections of these areas to posterior areas involved in stimulus representation (*Petrides, 2005*). Furthermore, we assumed domain-segregation such that spatial stimuli arrived at SFS, and verbal stimuli arrived at IFJ (*Goldman-Rakic, 1987*; *Romanski, 2004*), which was consistent with our univariate results demonstrating that these areas were the most active for those respective conditions. Cues denoting the appropriate sub-task were set as inputs into MFG consistent with its role in representing task rules (*Nee and Brown, 2012*; *Bunge et al., 2005*; *Genovesio et al., 2005*; *Wallis et al., 2001*). This assumption was bolstered by a univariate analysis demonstrating that MFG was the most active among the LPFC ROIs during the sub-tasks, and also by preliminary DCM's that directly compared models assuming cue input to MFG versus FPl, which demonstrated superior model fits for the former across a range of other parameters. Inputs were modeled as impulse events occurring at the onset of each stimulus.

Of central interest in DCM is how connectivity is modulated as a function of experimental manipulation. To investigate this matter, we used Bayesian model selection to adjudicate between different models positing different forms of connectivity modulation (*Stephan et al., 2009*). We examined how experimental manipulations of *Stimulus Domain*, *Contextual Control*, and *Temporal Control* modulated connectivity among the LPFC. *Stimulus Domain* was split separately into verbal and spatial factors. Modulations were explored during the sub-task phases of each block, modeling the same periods as the univariate analyses described above. We restricted exploration of modulations among areas showing a univariate effect of experimental manipulation (*Friston et al., 2003*). Explicitly, *Spatial Stimulus Domain* between SFS and cMFG, *Verbal Stimulus Domain* between IFJ and IFS, *Temporal Control* between FPl and MFG, and *Contextual Control* between all connected areas. We also explored lateral connectivity in the case of *Stimulus Domain* (IFJ-SFS, IFS-cMFG) to examine the possibility of inhibitory segregation. In different models, we estimated parameters assuming top-down, bottom-up, bi-directional, or no modulation. In this framework, there are over 20,000 potential models. We estimated that exhaustive exploration of the model space would take years of computational time. To make the problem computationally tractable, we reduced the model space by yoking the directionality of dorsal and ventral pathway modulations (e.g. if *Spatial Stimulus Domain* was a bottom-up modulator, then *Verbal Stimulus Domain* was also a bottom-up modulator for a particular model). Furthermore, we did not explore disjointed modulation in the case of the *Contextual Control* factor (i.e. modulations of rostral and caudal connections, but not mid-level connections). Finally, we used an iterative approach to traverse the remaining model space. First, we found the best model for modulations by *Stimulus Domain* (Model S). Next, assuming S, we found the best model for modulations by *Temporal Control* (Model T). Next, assuming S and T, we found the best model for modulations by *Contextual Control* (Model C). Thereafter, we revisited *Stimulus Domain* modulations assuming T and C (Model S'), then commensurately revisited *Temporal Control* assuming S' and C, and so on. We iterated this process until convergence. In each phase, the best model was chosen through Bayesian model selection as the model showing the highest model exceedance probability assuming random effects (*Stephan et al., 2009*). The model exceedance probability is the probability of one model being more likely than any other model under

consideration. The random effects procedure allows for the possibility of different models/strategies across participants and is far more robust against outliers than fixed effects procedures. Across the 6 iterations of the iterative procedure, the mean model exceedance probability of the best model was 0.72 (range 0.54–0.86). A total of 99 models were explored across the iterations. Comparing across the 99 tested models, the model that the iterative approach settled upon had a model exceedance probability of 0.18 (compare to a uniform of 0.01). This was more than twice the model exceedance probability of the next best model. Only 5 models had a model exceedance probability over 0.05. All of these 5 models were highly similar to one another (*Figure 6—figure supplement 5*). Hence, it is unlikely that the appropriate model for the observed dynamics is far different from the one explored in the text.

Random effects inference on parameters was performed on the winning model. These parameters included the strength of fixed connectivity (A matrix), input sensitivity (C matrix), and the strength of connectivity modulations (B matrix). P-values were corrected for multiple comparisons using false-discovery rate. To determine the robustness of these parameters, we split the dataset in two using an alternating runs procedure, excluding the participant that completed only a single session (*Figure 6—figure supplement 2*). To examine the relationship between top-down modulations, bottom-up modulations, and higher-level cognitive ability, we created factors for each metric using principle components analysis. The first eigenvariate was computed across all top-down modulations, all bottom-up modulations, and all trait-measures of cognitive ability, respectively. Signs were adjusted, if needed, so that higher positive values were associated with more of a given measure. These factors were then inter-correlated. To examine hierarchical strength, we contrasted the average efferent connectivity of a region with the average afferent connectivity along the rostral/caudal axis (connections running dorsal/ventral were ignored). This approach is similar to that used in a recent study involving hierarchical projections in the monkey (*Goulas et al., 2014*). Since the meaning of negative connectivity is unclear in this procedure, negative connections were set to 0 (17% of all connections across participants). The results remain if we omit this assumption. To provide a summary measure of hierarchical strength of mid LPFC, we fit a quadratic function to each participants' hierarchical strength metrics. In these models, the y-coordinate of each ROI, reflecting rostral/caudal location, was used as the predictor variable, while hierarchical strength was used as the predicted variable. The vertex of the quadratic function corresponds to either the apex of the hierarchy (peak) or basin of the hierarchy (trough). Inferences were performed on the rostral/caudal position and height of the vertex. The height of the vertex was used as a summary of hierarchical strength of fixed connectivity, and was correlated with the higher-level cognitive ability factor described above. Correlations were supplemented by robust regression, which reduces the impact of high-leverage outliers.

## Acknowledgements

This research was supported by National Institute of Neurological Disorders and Stroke Grant F32 NS0802069 (DN) and P01 NS040813 (MD), and National Institute of Mental Health Grant R01 MH063901 (MD). The authors thank Jack Gallant for helpful ideas on data analysis, and Max Wang and Lara Krisst for help with data collection.

## Additional information

### Funding

| Funder | Grant reference number | Author |
| --- | --- | --- |
| National Institute of Neurological Disorders and Stroke | F32 NS0802069 | Derek Evan Nee |
| National Institute of Mental Health | R01 MH063901 | Mark D'Esposito |
| National Institute of Neurological Disorders and Stroke | P01 NS040813 | Mark D'Esposito |

The funders had no role in study design, data collection and interpretation, or the decision to submit the work for publication.

**Author contributions**
DEN, Conception and design, Acquisition of data, Analysis and interpretation of data, Drafting or revising the article; MD, Conception and design, Drafting or revising the article

**Author ORCIDs**
Derek Evan Nee, http://orcid.org/0000-0001-7858-6871

**Ethics**
Human subjects: Informed consent was obtained in accordance with the Committee for Protection of Human Subjects at the University of California, Berkeley.

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
