## [Decision Letter]

Thank you for submitting your work entitled "The Hierarchical Organization of the Lateral Prefrontal Cortex" for consideration by *eLife*. Your article has been favorably evaluated by Timothy Behrens (Senior Editor) and three reviewers, one of whom, Lila Davachi, is a member of our Board of Reviewing Editors. The reviewer Charan Ranganath has agreed to reveal his identity.

The reviewers have discussed the reviews with one another and the Reviewing Editor has drafted this decision to help you prepare a revised submission.

Summary:

Overall, all three reviewers found the work novel and potentially very important as the conclusion put forth would challenge existing notions of the functional architecture of the prefrontal cortex. The analytic approach is impressive and sound. However, all three reviewers agreed that manuscript is very dense and difficult to unpack – and this will ultimately limit the impact of the paper. Also, there are some questions about the specificity of the brain/behavior correlations that should also be addressed.

Thus, it is critical that the manuscript be streamlined to focus on and clearly address the following:

1) The task – its elements, what cognitive control demands are being tested and how they relate to existing theories of cognitive control.

2) Consider limiting the analyses presented to address the main issue of what PFC regions are processing current as well as future demands. This seems like one fundamental contribution of the experimental design and results but this point should be highlighted and carried throughout the paper more clearly.

3) Please address the issues of selectivity with respect to the brain behavior correlations reported in Figure 8. Have the same analyses been computed with caudal and rostral PFC? Are those also significantly related to behavior? Is the noted relationship between mid-PFC hierarchical strength and behavior significantly stronger than that measured for caudal or rostral PFC?

4) Provide a clear description of DCM that includes consideration of what it can tell you as well as its limitations.

---

## [Author Response]

*Overall, all three reviewers found the work novel and potentially very important as the conclusion put forth would challenge existing notions of the functional architecture of the prefrontal cortex. The analytic approach is impressive and sound. However, all three reviewers agreed that manuscript is very dense and difficult to unpack* –

*and this will ultimately limit the impact of the paper. Also, there are some questions about the specificity of the brain/behavior correlations that should also be addressed. Thus, it is critical that the manuscript be streamlined to focus on and clearly address the following: 1) The task – its elements, what cognitive control demands are being tested and how they relate to existing theories of cognitive control.*

We recognize that the complexity of the task and terminology used makes it somewhat difficult to penetrate the operations involved and how they relate to the broader literature on cognitive control. We have taken several measures to address this issue. First, we have added a section at the beginning of the Results to better describe the task, the underlying cognitive control processes, and their relationship to the examined contrasts. Second, we have changed the names of the cognitive control processes to make better contact with previous literature. You will now find “Delayed Integration” referred to as “Temporal Control” and “Contextualization” as “Contextual Control.” We further spell-out how the statistical interaction of the cognitive control processes is expected to reveal processing related to selective attention for working memory, which we refer to as “Feature Control.” Finally, we have added a section in the Discussion describing theories and controversies about processing along the rostral/caudal axis of the LPFC and how the present data fit within that debate. We hope that these measures improve the readability of the manuscript and help situate the present study within the broader literature.

Added to the Results:

“To engage the full extent of LPFC necessary to test our predictions, we adapted a task (Koechlin et al., 1999; Charron and Koechlin, 2010) that orthogonally manipulated demands on stimulus domain (i.e. verbal vs. spatial) and cognitive control (Figure 1). […] Collectively, these manipulations were designed to engage the entire LPFC, providing an opportunity to examine the hypothesized hierarchical interactions among LPFC areas that support cognitive control. We focus our analyses on the sub-task trials, unless otherwise specified.”

Added to the Discussion:

“How to operationally define the rostral/caudal axis of the LPFC has been a matter of considerable debate. Gradients of activation along the rostral/caudal axis of the LPFC have been observed by manipulating the temporal timescale of control processing and relational complexity (see Badre, 2008 for an in-depth review). […] Hence, the temporal timescale of processing/representation increased from caudal to rostral areas.”

*2) Consider limiting the analyses presented to address the main issue of what PFC regions are processing current as well as future demands. This seems one fundamental contribution of the experimental design and results but this point should be highlighted and carried throughout the paper more clearly.*

We realize that by attempting to describe both the dorsal/ventral and rostral/caudal axes of the LPFC, the manuscript became dense and potentially unfocused. Since the rostral/caudal axis is of primary importance to the present work, we have removed much of the subject matter regarding the role and controversies surrounding the dorsal/ventral axis. We have retained analyses regarding stimulus domain-specificity/generality because we feel they are an important stepping-stone for demonstrating that the rostral/caudal axis shows an abstraction gradient. However, further focus on the dorsal/ventral axis has been attenuated. We hope that this improves the readability of the manuscript.

*3) Please address the issues of selectivity with respect to the brain behavior correlations reported in Figure 8. Have the same analyses been computed with caudal and rostral PFC? Are those also significantly related to behavior? Is the noted relationship between mid-PFC hierarchical strength and behavior significantly stronger than that measured for caudal or rostral PFC?*

Selectivity is an important matter and we are cognizant of the limitations of making associations without dissociations. However, in this matter, selectivity is a difficult matter to disentangle. Note that the measures of hierarchical strength are a contrast of outward relative to inward connectivity with respect to areas within the LPFC. By the nature of the contrasts, any two areas that are connected have dependencies on this metric – that is, hierarchical strength of connected areas is anti-correlated. As might be anticipated by such interdependencies, not only is hierarchical strength of the mid LPFC positively related to cognitive ability, caudal and rostral LPFC are negatively related to cognitive ability. This is due to the negative relationship between mid LPFC and both caudal and rostral LPFC that naturally arises due to the way in which hierarchical strength is calculated. Partial correlations were suggestive that the mid LFPC is the main driver, but not conclusive due to a lack of significance:

“The height of the vertex summarizes the hierarchical strength of mid LPFC. However, it is obtained by fitting the hierarchical strengths of all areas, thereby incorporating some information from other regions, as well. […] However, that the relationships between caudal/rostral LPFC and higher-level cognitive ability were negligible after partialing out the contributions of mid LPFC and that the reverse was not true is suggestive that the main driver is the hierarchical strength of the mid LPFC.”

*4) Provide a clear description of DCM that includes consideration of what it can tell you as well as its limitations.*

We recognize that the complexity of DCM can render it inaccessible. We have performed several measures to attempt to alleviate the issue. First, we have added a paragraph to the Results to conceptually describe its principles in more detail and support for its assumptions, along with its limitations (see below). Second, we have added a supplemental figure to provide a better sense of what aspects of the task are going into the DCMs (Figure 5—figure supplement 2). Third, we have added a supplemental figure to illustrate the features we are extracting from DCM to perform correlations with higher-level cognitive ability (Figure 8—figure supplement 2). We hope that this combination facilitates understanding and interpretation.

Added to the Results:

“Hierarchical interactions are presumed to be reflected by asymmetries in the influence of one region upon another (Badre and D'Esposito, 2009). […] This requires specifying multiple (plausible) models and comparing model evidence to reveal which model provides the best account of the data. Hence, DCM can only make relative claims with respect to the explored model space.”